

# Taxonomic reassessment of the genus *Dichotomius* (Coleoptera: Scarabaeinae) through integrative taxonomy

Carolina Pardo-Diaz[1], Alejandro Lopera Toro[2], Sergio Andrés Peña Tovar[3], Rodrigo Sarmiento-Garcés[4], Melissa Sanchez Herrera[1] and Camilo Salazar[1]

[1] Biology Program, Faculty of Natural Sciences and Mathematics, Universidad del Rosario, Bogota, D.C., Colombia
[2] Fundacion Ecotropico, Bogota, D.C., Colombia
[3] Universidad Distrital Francisco José de Caldas, Bogota, D.C., Colombia
[4] Facultad de Ciencias, Universidad Nacional de Colombia, Bogota, D.C., Colombia

## ABSTRACT

Dung beetles of the subfamily Scarabaeinae are widely recognised as important providers of multiple ecosystem services and are currently experiencing revisions that have improved our understanding of higher-level relationships in the subfamily. However, the study of phylogenetic relationships at the level of genus or species is still lagging behind. In this study we investigated the New World beetle genus *Dichotomius,* one of the richest within the New World Scarabaeinae, using the most comprehensive molecular and morphological dataset for the genus to date (in terms of number of species and individuals). Besides evaluating phylogenetic relationships, we also assessed species delimitation through a novel Bayesian approach (iBPP) that enables morphological and molecular data to be combined. Our findings support the monophyly of the genus *Dichotomius* but not that of the subgenera *Selenocopris* and *Dichotomius* sensu stricto (s.s). Also, our results do not support the recent synonymy of *Selenocopris* with *Luederwaldtinia*. Some species-groups within the genus were recovered, and seem associated with elevational distribution. Our species delimitation analyses were largely congruent irrespective of the set of parameters applied, but the most robust results were obtained when molecular and morphological data were combined. Although our current sampling and analyses were not powerful enough to make definite interpretations on the validity of all species evaluated, we can confidently recognise *D. nisus, D. belus* and *D. mamillatus* as valid and well differentiated species. Overall, our study provides new insights into the phylogenetic relationships and classification of dung beetles and has broad implications for their systematics and evolutionary analyses.

## INTRODUCTION

Scarabaeinae dung beetles are one of the most morphologically diverse groups of animals (*Philips, 2011*) comprising more than 6,000 species and 200 genera worldwide (*Tarasov & Génier, 2015*). Within this dung-feeding subfamily, *Dichotomius Hope, 1838* constitutes one

Corresponding author
Carolina Pardo-Diaz,
geimy.pardo@urosario.edu.co,
cabardia@gmail.com

of the richest genera endemic to the Americas, with 171 described species (*Schoolmeesters, 2019*). Compared to other regions, its diversity is highest in South America where more than 100 species can be found (*Bohórquez & Montoya, 2009*; *Vulinec, 1999*). Species in this genus vary in size (5–38 mm), show strong sexual dimorphism and have colours usually ranging from dark brown to black (*Nunes, 2017*; *Sarmiento-Garcés & Amat-García, 2014*; *Vaz-de-Mello et al., 2011*). Furthermore, *Dichotomius* species are typically nocturnal, more abundant in the rainy season and prevalent in several Neotropical terrestrial habitats where they play multiple ecological roles (*López-Guerrero, 2005*; *Maldaner, Nunes & Vaz-De-Mello, 2015*; *Vulinec, 1999*). For example, they promote bioturbation, remove faeces from forests and pastures, bury seeds, stimulate seed germination and even act as intermediate hosts of swine parasites (*Almeida et al., 2014*; *Nichols et al., 2008*; *Vulinec, 1999*).

The taxonomy of these beetles, which is entirely based on morphological characters, is still not sufficiently resolved despite them being ubiquitous and ecologically relevant. The genus was divided into four subgenera by *Luederwaldt (1929)*: *Dichotomius* sensu stricto (s.s.), *Selenocopris*, *Homocanthonides* and *Cephagonus* (*Luederwaldt, 1929*). Since then there have been few changes, the most relevant done by *Martínez (1951)* that keeps *Dichotomius* s.s. and *Homocanthonides*, but changes *Selenocopris* to *Luederwaldtinia* and *Cephagonus* to *Selenocopris* (*Martínez, 1951*). The most recent revision of the genus *Dichotomius* differentiates the four subgenera based mainly on variations of the clypeo-genal angle (*Nunes, 2017*) supporting the initial division by Luederwaldt in 1921: *Dichotomius* s.s. (70 spp); *Homocanthonides* (1 spp); *Cephagonus* (16 spp) and *Selenocopris* (75 spp), synonymising the latter with *Luederwaldtinia*. These subgenera are further divided into species groups, each one containing multiple species (*Luederwaldt, 1929*; *Martínez, 1951*; *Nunes, 2017*; *Nunes & Vaz-de-Mello, 2013*; *Nunes & Vaz-de-Mello, 2016*). Although there has been a recent interest in revising these subgenera and species groups, their definition is still problematic due to relying on morphological traits alone (*Maldaner, Nunes & Vaz-De-Mello, 2015*; *Nunes, 2017*; *Nunes & Vaz-de-Mello, 2013*; *Nunes & Vaz-de-Mello, 2016*). This problem also applies to species delimitation in the genus because some species such as *Dichotomius satanas* display a spectacular range of morphological variability, which suggests the possibility of distinct species being misclassified as a single one (*Sarmiento-Garcés & Amat-García, 2014*). In fact, some authors consider *D. satanas* as a species complex in need of revision (*Nunes, 2017*). For example, specimens of *D. satanas* from Central America have been reported to look different from those from Colombia (with the type being from this country), and within Colombia, females of *D. satanas* from the Eastern Cordillera have two or four protuberances on the pronotum while females from the Western and Central cordillera have only two (Fig. S1) (*Sarmiento-Garcés & Amat-García, 2014*).

The use of molecular tools constitutes an alternative to accurately delimit and identify taxa that lack useful morphological characters (*Dayrat, 2005*; *Dupuis, Roe & Sperling, 2012*; *Schlick-Steiner et al., 2009*; *Schwarzfeld & Sperling, 2014*). This approach has been primarily used in Scarabaeinae beetles to resolve deep relationships (*Gunter et al., 2016*; *Tarasov & Génier, 2015*); however, the relationships at the genus or species level in this subfamily remains understudied. For this reason, there is currently no molecular phylogeny available for *Dichotomius.* Recent studies on deep phylogenies for Coleoptera and dung beetles,

however, indicate that the genus is likely paraphyletic (although this result is based on a small number of species of *Dichotomius* and only one individual per species) (*Bocak et al., 2014*; *Monaghan et al., 2007*).

In recent years taxonomists have begun to integrate different lines of evidence to discover and delimit species, which is often referred to as ''integrative taxonomy'' (*Padial & De La Riva, 2010*; *Schlick-Steiner et al., 2009*). The application of this approach, usually done through the combination of molecular and morphological information, has improved taxonomic rigor yielding a more precise biodiversity inventory (both reducing or increasing species numbers) (*Sturaro et al., 2018*). In this study, we implemented an integrative taxonomy approach that combines morphological and molecular data (both mitochondrial and nuclear) to make a preliminary assessment of the species diversity and phylogenetic relationships in the genus *Dichotomius*. The information derived from this research is crucial to further characterise species' richness as well as to understand patterns of adaptation, speciation and biogeography in these dung beetles.

## MATERIALS & METHODS

### Sampling

Our total sample set consisted of 304 individuals of *Dichotomius* (31 species). The morphological analysis of male genitalia included 208 individuals from 28 species (Table S1), whereas the genetic analysis consisted of 145 specimens from 16 species; 52 of these sequences were obtained from GenBank (Table S1). This is representative of 14 species-groups and three subgenera in *Dichotomius*. All specimens for which we obtained data (DNA or morphology) came from the following biological collections: (i) Colección Alejandro Lopera-Toro (CALT-ECC, Colombian Collection ID 2), (ii) Museo de Historia Natural Universidad Distrital (MUD, Colombian Collection ID 46), and (iii) Colección de Artrópodos de la Universidad del Rosario (CAUR, Colombian Collection ID 229). These individuals were identified by experts or using most recent taxonomical keys (*Nunes, 2017*; *Sarmiento-Garcés & Amat-García, 2014*; *Vaz-de-Mello et al., 2011*).

### Morphometric analyses

Because the shape of the male genitalia is considered one of the most informative morphological characters in the classification of *Dichotomius* species (*López-Guerrero, 2005*; *Sarmiento-Garcés & Amat-García, 2014*), we analysed the quantitative variation of the aedeagus in 208 individuals (28 species; Table S1). Male genitalia preparation followed a standard procedure: we detached the last two abdominal segments, soaked them in 10% KOH at 60 °C–70 °C for 12 h and neutralized them in 1% acetic acid to finally store them in glycerine (*Sarmiento-Garcés & Amat-García, 2014*). Then, we cleaned and dissected the aedeagus. Finally, we photographed the aedeagus in dorsal view and using a Leica DFC320 digital camera coupled to a Leica S6 stereoscope at 4X magnification.

We applied landmark-based geometric morphometrics to these photographs in order to analyse genital shape. We used tpsDig v.2.31 (*Rohlf, 2004*) to digitise 33 landmarks per individual that describe the outline of the aedeagus, all of them were placed on the parameres (Fig. S2A). This landmark dataset was subjected to superimposition using a

Generalized Procrustes Analysis (GPA) in the R package 'geomorph' (*Adams & Otárola-Castillo, 2013*). For this, the software aligns, scales and rotates the configurations to line up the corresponding landmarks as closely as possible, minimizing differences between landmark configurations without altering shape. Then, we obtained partial warps (or shape variables) that indicate partial contributions of hierarchically scaled vectors spanning a linear shaped space. With this information we generated a consensus shape that summarises the aedeagus' shape variation among all *Dichotomius* species included (Fig. S3). In this way, each specimen's shape is quantified by the deviation of its landmark configuration from the average landmark configuration (i.e., consensus shape), which allows to visualise differences between groups. Differences in aedeagus' shape among species were tested using a Procrustes MANOVA applied to the aligned landmark configurations. This was done using the *procD.lm* function in the 'geomorph' R package (*Adams & Otárola-Castillo, 2013*).

We implemented a principal component analysis (PCA) on the procrustes aligned data using the *plotTangentSpace* function in the 'geomorph' R package (*Adams & Otárola-Castillo, 2013*). Of the 66 PCs produced, the first two cumulatively accounted for ~92% of the total shape variance; therefore, further analyses were performed on these PCs. We used the function *plotRefToTarget* from the same package to generate the deformation grids representing the extremes (maximum and minimum) of shape variation along the principal components 1 and 2 (PC1 and PC2). We then applied a discriminant analysis of principal components (DAPC) using the R package 'adegenet' (*Jombart, 2008*).

We also applied a model-based hierarchical clustering using the R package 'mclust' (*Scrucca et al., 2016*) in order to identify groups of individuals that resemble each other, independent of other evidence or *a priori* assignments. This method uses expectation maximization (EM) to estimate the Maximum Likelihood (ML) of alternative multivariate mixture models that describe shape variation in the data and estimates the optimal number of clusters based on the Bayesian Information Criterion (BIC). All models were evaluated for a predefined number of 1 to the maximum number of morphospecies studied (28 in our case, i.e., those for which morphology data was available).

## Molecular analyses

We extracted DNA from legs of 95 specimens of *Dichotomius* using the DNeasy Blood & Tissue Kit (QIAGEN, Hilden, Germany) following the manufacturer's instructions with minor modifications: 40 μL of proteinase K were used, protein digestion lasted for at least 2 h and the final elution was made in 100 μL of warm AE buffer. Then, we amplified the 3′ and 5′ ends of the cytochrome c oxidase I gene (*COI*), and the nuclear gene 28S. All PCR reactions were performed in a final volume of 10 μL containing one μL of 10X Buffer, 0.6 μL of $MgCl_2$ (25 mM), 0.5 μL of dNTP mix (10 mM), 0.5 μL of each primer (10 μM), 0.05 μL of DNA polymerase (five U/μl; QIAGEN) and 5.85 μL of $dH_2O$. To amplify the 3′ end of the COI gene we used the primers C1-J-2183 (Jerry: 5′-CAACATTTATTTTGATTTTTTGG-3′) and TL2-N-3014 (Pat: 5′-TCCAATGCACTAATCTGCCATATTA-3′) (*Simon et al., 1994*). The amplification PCR profile consisted of an initial denaturation step of 94 °C for 5 min, 7 cycles of denaturation at 94 °C for 1 min, annealing at 48 °C for 45 s and

extension at 72 °C for 1 min, followed by 33 cycles of denaturation at 94 °C for 45 s, annealing at 52 °C for 45 s and extension at 72 °C for 1.5 min, with a final extension at 72 °C for 10 min. The 5′ end of the COI gene (the barcode) was amplified with the primers LCO1490 (5′-GGTCAACAAATCATAAAGATATTGG-3′) and HCO2198 (5′-TAAACTTCAGGGTGACCAAAAAATCA-3′) (*Folmer et al., 1994*), using the following PCR conditions: 94 °C for 5 min, 35 cycles of 94 °C for 30 s, 45 °C for 30 s, 72 °C for 1.5 min and a final extension at 72 °C for 7 min. To amplify the 28S gene we used the primers 28SFF (5′-TTACACACTCCTTAGCGGAT-3′) and 28SDD (5′-GGGACCCGTCTTGAAACAC-3′) (*Monaghan et al., 2007*). PCR cycling was 94 °C for 5 min, 38 cycles of 94 °C for 30 s, 53 °C for 30 s, 72 °C for 45 s and a final extension of 72 °C for 10 min.

All PCR products were purified with ExoSAP and their bidirectional sequencing was carried out by ELIM Biopharmaceuticals Inc. (Hayward, CA, USA). Forward and reverse sequences from each amplicon were verified and assembled into a single consensus contig based on a minimum match of 80% and a minimum overlap of 50 bp using CLC main workbench.

Sequences of each genetic marker were aligned independently using MUSCLE (*Edgar, 2004*) in MESQUITE v3.04 (*Maddison & Maddison, 2011*); poorly aligned regions were corrected manually. Protein coding sequences were translated into amino acids to confirm the absence of stop codons and anomalous residues in MESQUITE v3.04 (*Maddison & Maddison, 2011*). Additional sequences of *Dichotomius* available in GenBank (Table S1) were downloaded and integrated into the alignments. All sequences generated by us were deposited in GenBank and their accession numbers are listed in Table S1.

We estimated a phylogenetic tree based on the sequence information from the COI, 28S and 16S. All sequences from the latter marker were obtained from GenBank and correspond to the species *D. nisus*, *D. yucatanus*, *D. parcepunctatus* and *D. boreus* (Table S1). We concatenated all genes into a single alignment (2,546 bp) that included 16 species of *Dichotomius* and nine outgroups: *Deltochilum larseni*, *Neateuchus proboscideus*, *Ontherus diabolicus*, *Pedaria* sp., *Panelus* sp., *Australammoecius occidentalis*, *Euphoresia* sp., *Brindalus porcicollis*, *Pleurophorus caesus* (Table S1). We calculated a ML tree using IQ-TREE using the entire haplotype set derived from all species and individuals (*Nguyen et al., 2015*) with 1,000 ultrafast bootstrap replicates. This was done based on the substitution model showing the smallest AIC score for each partition (i.e., COI, 28S and 16S), which was also selected using IQ-TREE (*Nguyen et al., 2015*; Table S2).

To test whether *D. satanas* exhibits genetic clustering associated to the Colombian Cordilleras of the Andes as previously suggested (*Sarmiento-Garcés & Amat-García, 2014*), we also estimated a ML topology using all sequences available for the Colombian specimens of this species (COI and 28S) and using the conditions aforementioned. The sequences were all concatenated into a single alignment of 2,145bp that included one individual of *D. boreus, D. quinquelobatus* and *D. protectus* (outgroups) and 60 individuals of *D. satanas:* 8 from the Central Cordillera of Colombia, 14 from the West Cordillera of Colombia and 38 from the East Cordillera of Colombia.

Finally, we used DnaSP version 6.12.01 (*Rozas et al., 2003*) to calculate diversity parameters (i.e., number of haplotypes (H), haplotype diversity, genetic diversity ($\pi$

and $\theta$) and Tajima's D) for all species and for *D. satanas,* as well as summary statistics of population differentiation among populations of *D. satanas*.

## Species delimitation analyses

We implemented a joint Bayesian inference based on genetic and phenotypic data to delimit species using iBPP (*Solís-Lemus, Knowles & Ané, 2014*). This was done using two independent data sets: (i) all species, and (ii) *D. satanas* from Colombia only. In both cases, we ran the program for three different datasets: (i) morphological and molecular data combined, (ii) morphological data alone, (iii) molecular data alone. In all cases, we used the species-tree topology from IQ-tree as the guide tree. The morphological character matrix used as input included the values of PC1 and PC2 from the geometric morphometric analyses. The molecular matrix included all sequences available for the markers COI, 16S and 28S. We specified nine combinations of the prior distribution for the ancestral population size ($\theta$) and the root age of the tree ($\tau$) ranging from scenarios that represent large population sizes and a deep divergence time ($\theta = G(1, 10)$ and $\tau = G(1, 10)$) to those representing small population sizes and a shallow divergence time ($\theta = G(2, 2000)$ and $\tau = G(2, 2000)$) as previously used (*Eberle, Warnock & Ahrens, 2016*; *Olave et al., 2017*). We used default values of $\sigma^2$ and $\kappa = 0$, thus these priors are non-informative and the program estimates them. The MCMC analysis was run over 50,000 generations, sampling every 1,000 steps and using a 10% burn-in. We confirmed the robustness of the results by running the analysis with both the algorithms 0 and 1 for rjMCMC searches. As results were very similar, we present those of algorithm 1. The parameters of the locus-specific rates of evolution were fine-tuned using an auto option.

## RESULTS

### Morphological analyses

When we tested for aedeagus shape variation in the entire Procrustes shape space, we found differences among all categories tested (i.e., subgenera, species-groups and species; Procrustes MANOVA $p < 0.001$ in all cases). The PCA of the aedeagus shape revealed that most of its variation is contained in few dimensions. The first two PCs accounting for 91.9% of the total variance. PC1 explained 84.16% of the aedeagus shape and was driven by the width of the lateral outer margins in the apex of the parameres, ranging from being broad to narrow (Fig. 1A and Fig. S2). PC2 explained 7.7% of morphological of the aedeagus shape variation and describes the shape formed by the sides of the parameres (Fig. 1A and Fig. S2). The DAPC suggests the existence of four discrete genitalia morphology groups within *Dichotomius* (Fig. 1B and Fig. S4). The first group (depicted in red tones) was composed mostly by members of the subgenus *Selenocopris* sensu (*Nunes, 2017*) from the species-groups Agenor, Batesi and Inachus (i.e., *D. agenor, D. batesi, D. belus, D. deyrollei, D. favi, D. fortestriatus,* and *D. yucatanus*). This group also contained individuals of the subgenus *Dichotomius* s.s., exclusively those in the species-group Carolinus (i.e., *D. amicitiae* and *D. coenosus*). Finally, the species *D. fonsecae* (subgenus *Cephagonus,* species group Fissus) also clustered in this first group. The second group (depicted in green tones) was mainly formed by species that belong to the subgenus

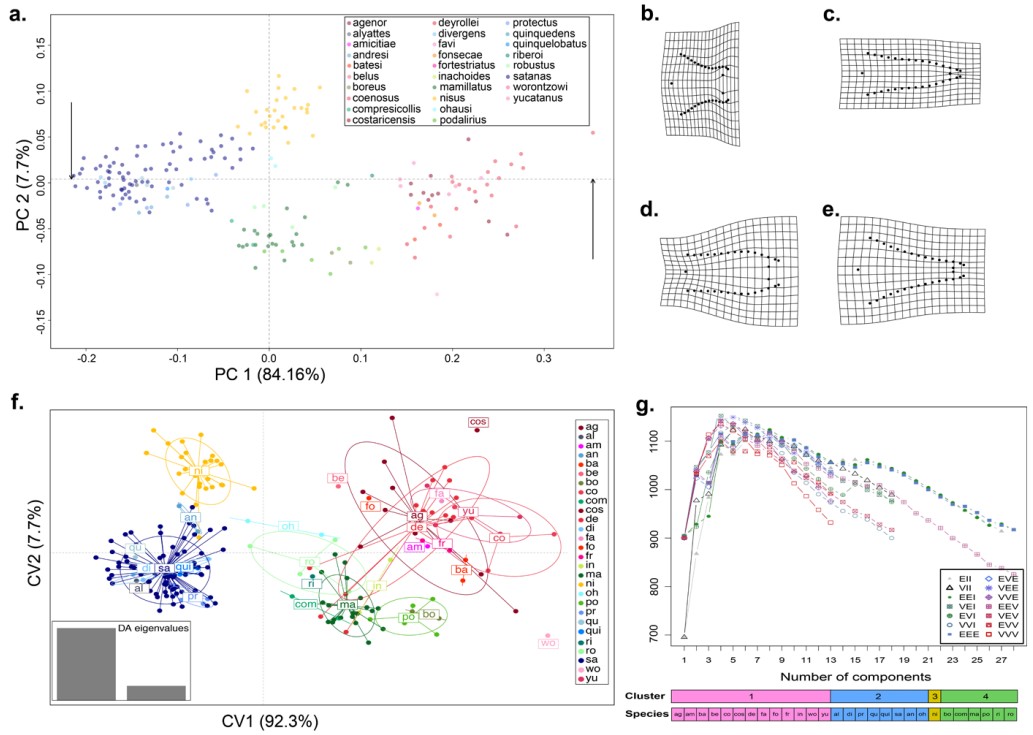

**Figure 1 Shape variation of the aedeagus in 28 species of *Dichotomius*.** (A) Principal Component Analysis (PCA). Deformation grids showing the maximum (B) and minimum (C) shape change of the aedeagus associated with PC1. Deformation grids showing the maximum (D) and minimum (E) shape change of the aedeagus associated with PC2. (F) Scatter plot of the DAPC analysis with species identity as prior information; ellipses correspond to the 95% confidence interval around the centroid. (G) Model based clustering showing the best fitting cluster model by BIC reassignment probabilities to the clusters with individuals ordered by cluster; bars below represent the reassignment probabilities to the clusters with individuals ordered by cluster and by *a priori* defined morphospecies. (ag, agenor; al, alyattes; am, amicitiae; an, andresi; ba, batesi; be, belus; bo, boreus; co, coenosus; com, compresicollis; cos, costaricensis; de, deyrrollei; di, divergens; fa, favi; fo, fonsecae; fr, fortestriatus; in, inachus; ma, mamillatus; ni, nisus; oh, ohausi; po, podalirius; pr, protectus; qu, quinquedens; qui, quinquelobatus; ri, riberoi; ro, robustus; sa, satanas; wo, worontzowi; yu, yucatanus).

*Dichotomius* s.s. from the species-groups Boreus, Buqueti and Mamillatus (i.e., *D. boreus, D. compresicollis, D. mamillatus, D. podalirius, D. riberoi* and *D. robustus*); the species *D. inachoides* (subgenus *Selenocopris*, species-group Agenor) also grouped here. The third group (yellow) consisted exclusively of individuals from *D. nisus* (isolated species in the *Selenocopris* subgenus sensu *Nunes, 2017*). The fourth group comprised only species from the subgenus *Dichotomius* s.s., species-group Mormon, namely: *D. alyattes, D. andresi, D. ohausi, D. protectus, D. divergens, D. quinquelobatus, D. quinquedens* and *D. satanas* (blue tones). Although the species *D. costaricensis* and *D. worontzowi* (both of the *Dichotomius* s.s. subgenus) appeared well differentiated from any other species or group, we only have one sample for each of them, preventing us from making strong inferences. Consistently, mclust identified four clusters entirely coincident with the groupings obtained above

(Fig. 1C). This variation is best explained by a model with 'diagonal distribution, variable volume and equal shape' (VEI; BIC = 1,152.184).

In summary, variation in genitalia morphology is not entirely consistent with the current taxonomy of *Dichotomius* (*Nunes, 2017*). Specifically, *D. (Selenocopris) nisus* (yellow) appears as different from other species in the subgenus *Selenocopris* (red). Also, species in the Carolinus group (*D. amicitiae* and *D. coenosus*), currently classified as members of *Dichotomius* s.s., cluster with species from the subgenus *Selenocopris* (red). Species in the subgenus *Dichotomius* s.s. formed two clusters, one that contains lowland species (green) and the other composed only by highland Andean species (blue).

## Molecular analyses

We found *Dichotomius* as a monophyletic genus with two well-supported deep clades (Fig. 2, Fig. S5). The first clade contains *D. (Selenocopris) nisus* sister to species from the subgenus *Dichotomius* s.s. The second clade is composed of species from the *Selenocopris* subgenus, except for *D. carolinus*, which is currently included within *Dichotomius* s.s. Within the first clade (1 in Fig. 2), all species that belong to *Dichotomius* s.s. were grouped by species-group, with the Mormon, Boreus and Mamillatus groups forming each a monophyletic cluster (Fig. 2; Fig. S5). Within each of these species-groups most species appeared as monophyletic, except for *D. satanas*. This species formed two monophyletic clades, one consisting of Colombian specimens and the other composed by Central American individuals (Fig. 2; Fig. S5). Within the second monophyletic clade (2 in Fig. 2) we could not test the monophyly of the species groups due to low species sampling, yet we observed the Agenor species-group as paraphyletic (Fig. 2; Fig. S5). In general, mtDNA showed higher haplotype diversity than the 28S nuclear gene (Table 1).

When populations of *D. satanas* from Colombia were analysed separately to evaluate whether this species displays genetic clustering associated with geography or phenotype (*Sarmiento-Garcés & Amat-García, 2014*), we mainly observed clustering and genetic differentiation associated to the three Cordilleras of the north of the Andes (Fig. 3, Table 2). Individuals from the Central and the Western Cordilleras were reciprocally monophyletic, and both were sister to the Western Cordillera clade. Interestingly, this phylogenetic pattern associates to morphological differences in the females: the Central and Western clusters contain females with only two protuberances in the pronotum, while the cluster of the Eastern Cordillera includes females with two and four protuberances. At the same time, the latter cluster separates into two inner groups, one that contains only females with four protuberances and the second, where females of two and four protuberances are found (Fig. 3).

## Species delimitation

The total-evidence (morphology and DNA) approach to Bayesian species delimitation (iBPP) did not support the *a priori* morphospecies assignment (Fig. 4). In most $\theta$ and $\tau$ scenarios tested, the posterior probability for the existence of the 16 morphospecies evaluated was lower than 50%. The only *a priori* defined species that consistently presented high support for all prior combinations were *D. belus, D. nisus* and *D. mamillatus.* Other

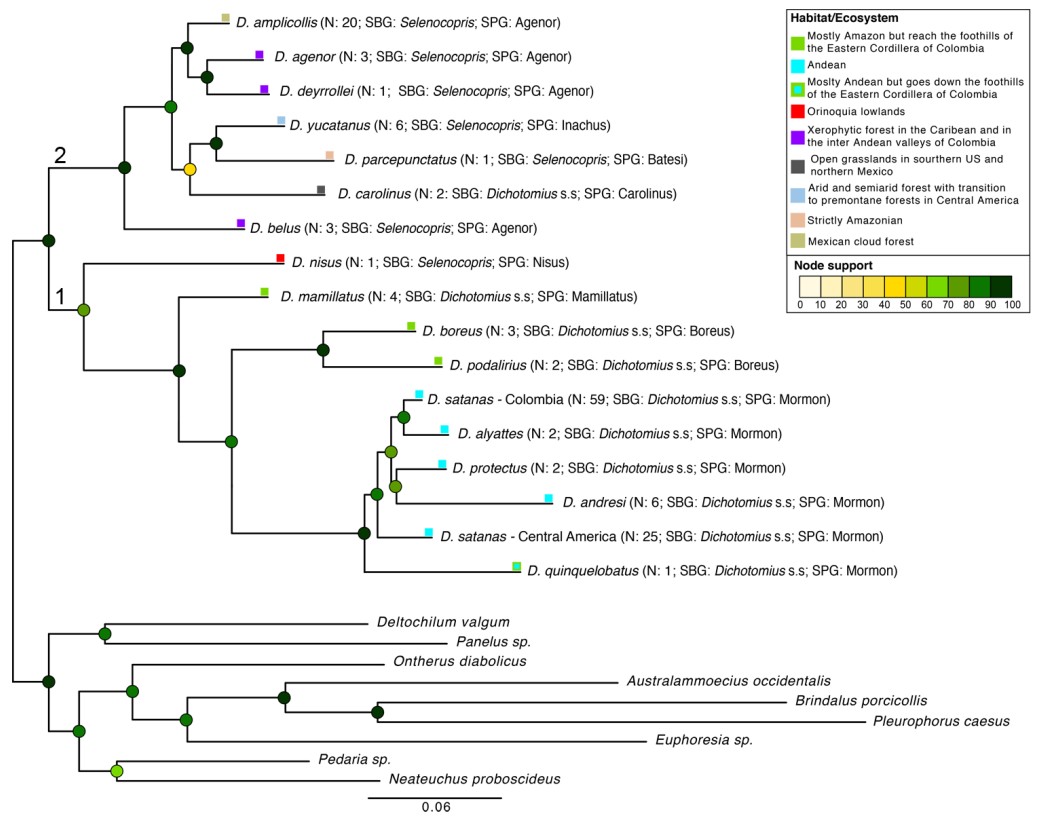

**Figure 2** **Phylogenetic relationships of *Dichotomius* species.** Summary phylogeny based on the ML tree of 16 species of *Dichotomius* and nine outgroup species (full phylogeny is shown in Fig. S5). Next to the species name we indicate the number of individuals within each collapsed branch (N), subgenus (SBG) and species group (SPG). Squares mapped onto branches indicate habitat/ecosystem. Circles on nodes indicate bootstrap support. Deepest clades are numbered as 1 and 2 as a reference in the main text.

**Table 1** **Genetic diversity indices for all species and for *D. satanas*.**

| Gen | | Number of haplotypes (H) | Haplotype diversity | Nucleotide diversity ($\pi$) | Substitution rate ($\theta$) | Tajima's D |
|---|---|---|---|---|---|---|
| COI | *D. satanas* | 29 | 0.95 | 0.02875 | 0.02455 | 0.4089 (ND) |
| | All species | 74 | 0.98 | 0.07645 | 0.06515 | 0.6996 (ND) |
| 28S | *D. satanas* | 3 | 0.59 | 0.00248 | 0.00229 | 0.268 (ND) |
| | All species | 10 | 0.84 | 0.02057 | 0.01225 | 2.249[*] |

**Notes.**
ND, non-different from zero.
[*]Significance < 0.05.

species were supported only when modelling small population sizes ($\theta = 0.01$) and medium to deep divergence time ($\tau = 0.05$ and $\tau = 0.1$), but never when modelling a shallow divergence time ($\tau = 0.01$; Fig. 4).

The existence of two deep clades was strongly supported, regardless of the $\theta$ and $\tau$ *priors* used (1 and 2 in Fig. 4). In the first clade, the existence of species groups Nisus,

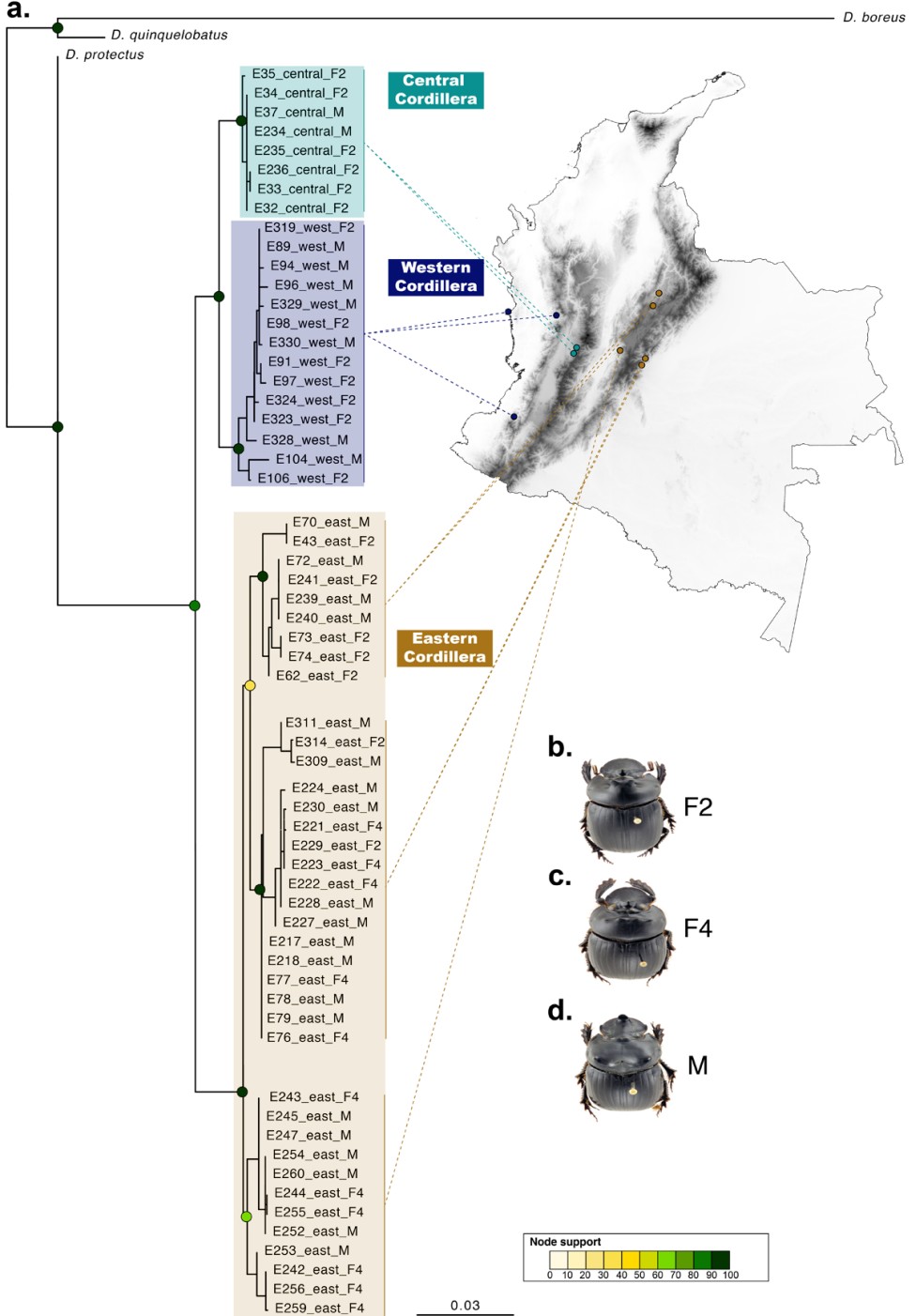

**Figure 3** **Phylogenetic relationships and phenotype variation in Colombian populations of *Dichotomius satanas*.** (A) ML tree based on the concatenation of the COI and 28S genes. Circles on nodes indicate bootstrap support. Coloured squares highlight geographic clusters and are connected to the collecting localities in Colombia. (B–D) Photos show the phenotype of males (M) and females, that can either have two (F2) or four (F4) protuberances in the pronotum.

**Table 2  Summary statistics of population differentiation among populations of *D. satanas*.**

| | WC—CC | | | WC—EC | | | CC—EC | | |
|---|---|---|---|---|---|---|---|---|---|
| | $N_{ST}$ | $D_{XY}$ | $D_a$ | $N_{ST}$ | $D_{XY}$ | $D_a$ | $N_{ST}$ | $D_{XY}$ | $D_a$ |
| COI | 0.34** | 0.04269 | 0.02173 | 0.19** | 0.04810 | 0.01000 | 0.51** | 0.03199 | 0.01536 |
| 28S | NA | 0.000001 | 0.000001 | 0.66** | 0.00332 | 0.00218 | 0.56** | 0.00382 | 0.00214 |

**Notes.**

WC, Western Cordillera; CC, Central Cordillera; EC, Eastern Cordillera.

Central America was not included because its sequences were only available for one fragment. NA, not computable.

**0.001 < $p$ < 0.01.

***$p$ < 0.001.

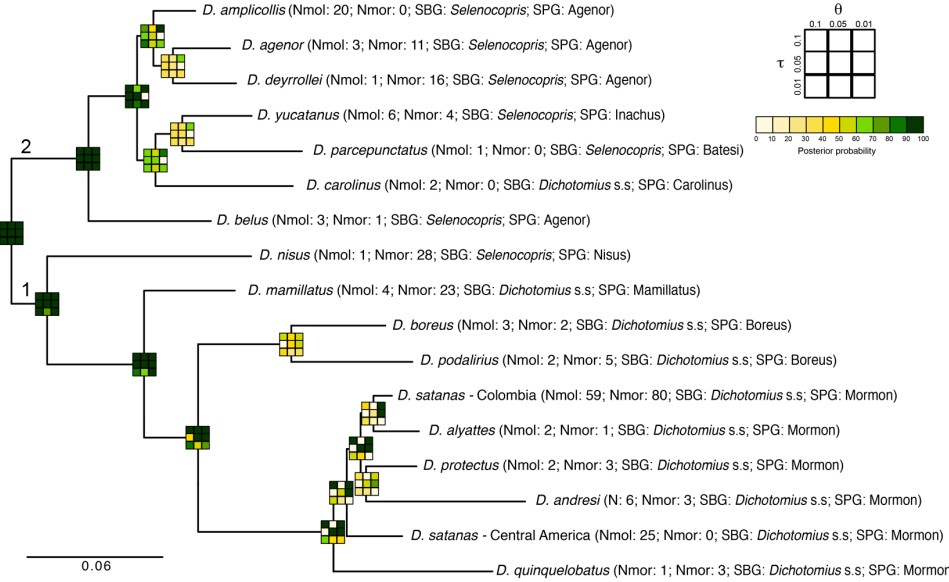

**Figure 4  Total-evidence Bayesian species delimitation.** Mean posterior probabilities of Bayesian species delimitations were inferred under nine different theta and tau *prior* combinations. The posterior probability of each of these combinations is colour-coded and indicated in 3 × 3 boxes on each node of the guide tree. The large 3 × 3 inset indicate the position of each prior combination in these boxes. Next to the species name we indicate the number of individuals included per species in the analysis (Nmol, number of individuals with molecular data; Nmor, number of individuals with morphological data). Subgenus (SBG) and species group (SPG) are also indicated. Deepest clades are numbered as 1 and 2 as a reference in the main text.

Mamillatus, Boreus and Mormon was also stronly supported (Fig. 4). In the latter group, the separation of *D. quinquelobatus* from other members of this group showed high posterior probability values in most scenarios, except for those with $\tau = 0.01$. However, the separation of *D. protectus* from *D. andresi,* or Colombian *D. satanas* from *D. alyattes* was rarely supported (Fig. 4). This was also observed in the Boreus species-group, where the delimitation between *D. boreus* and *D. podalirius* always had low posterior probabilities (Fig. 4).

In the second clade the only well supported species across all the parameter combinations ($\theta$ and $\tau$) was *D. belus*. The later suggest that this species may not be part of the Agenor

species group. In contrast, species within the clade sister to *D. belus* showed very low support in most of the parameter space (Fig. 4). The species delimitation based on molecular or morphological data alone were consistent with the total-evidence approach (Fig. S6). However, the results of these independent data types tended to provide stronger supports to species-groups and some species, especially the molecular data.

Finally, the total-evidence analysis of species delimitation done in *D. satanas* failed to identify any of the phylogenetic clusters associated to geography as separate species (in most $\theta$ and $\tau$ scenarios tested the support for these clusters was lower than 60%, Fig. S7A). This suggests that *D. satanas* is likely a single species with phenotypic polymorphism. However, just as before, the analyses with only molecular data presented stronger supports while the analysis based on morphological data provided very poor support Figs. S7B and S7C).

## DISCUSSION

Since the first description of *Dichotomius* by *Hope (1838)* about 170 species have been described in the genus using morphology as the only diagnostic tool, and although there have been recent morphological revisions, *Dichotomius* remains a challenging taxonomic puzzle (*Nunes, 2017*; *Nunes & Vaz-de-Mello, 2013*; *Nunes & Vaz-de-Mello, 2016*; *Nunes et al., 2012*). Here we used aedeagus morphology and phylogenetic analyses to assess the validity of some species in this dung beetle genus. Our study suggests it is necessary to make a comprehensive revision of the number of species within the genus that combines molecular and morphological data, as well as a broader taxonomic and geographic sampling.

Despite what previous deep phylogenies of the subfamily Scarabaeinae had suggested (*Bocak et al., 2014*; *Monaghan et al., 2007*), we found *Dichotomius* as a monophyletic genus. This is likely because our study is the first to include a more extensive sampling of species and individuals in this genus. We also showed that the subgenera *Dichotomius* s.s. and *Selenocopris* previously established by morphology (*Nunes, 2017*) were not supported. Regardless of the non-validity of these subgenera, our data recovered two well supported monophyletic clades consistent with distributional patterns, where species in clade one occur in both Central and South America, and species in clade two are restricted to South America with only one exception: *D. satanas* (Fig. 2).

The position of *D. nisus* outside *Selenocopris* and the inclusion of the Carolinus group inside this subgenus causes the non-monophyly of *Dichotomius* s.s. and *Selenocopris*. Until recently, *D. nisus* was recognised as the type species for the *Luederwaldtinia* subgenus (*Martínez, 1951*) but because both *Luederwaldtinia* and *Selenocopris* subgenera include described species that have clypeal teeth but lack clypeo-genal angle, Nunes synonymised *Luederwaldtinia* with *Selenocopris* (*Nunes, 2017*). Even so, Nunes still recognised *D. nisus* as unique within *Selenocopris*, leading to its classification in a separate species-group as an "isolated species" (*Nunes, 2017*; *Nunes & Vaz-de-Mello, 2013*). However, our data does not agree with this synonymisation as neither the aedeagus morphology nor the molecular data support the placing of *D. nisus* within *Selenocopris* and, in fact, both data types show this species more closely related to members of *Dichotomius* s.s. Also, *D. nisus* has a unique distribution and ecology that differentiates it from other *Dichotomius,* being a

common species that is restricted to Orinoquia lowlands, pastures and open environments (*França et al., 2016*; *Louzada & Carvalho E Silva, 2009*). Therefore, the resurrection of *Luederwaldtinia* with *D. nisus* as type species or its inclusion within *Dichotomius* s.s. needs to be evaluated by studying the morphology and DNA variation of all the species classified under both subgenera. On the other hand, the inclusion of the Carolinus species-group as part of *Selenocopris* would make this subgenus monophyletic, and makes sense in the light of geographic distribution, since the Carolinus species-group is restricted to Central America (where *Dichotomius* s.s. does not usually occur).

The Agenor species-group (i.e., *D. agenor, D. deyrollei, D. amplicollis* and *D. belus*) was not monophyletic since the molecular phylogeny and the total-evidence delimitation analysis strongly supported the exclusion of *D. belus* from it (Fig. 2). This separation may reflect differences in ecology or distribution of *D. belus* from the other members of the Agenor species-group. For instance, while all these species occur in xerophytic forests, *D. belus* is the only of them that can reach elevations up to 2,200 masl (*Arellano, León-Cortés & Halffter, 2008*; *Giraldo, Montoya & Escobar, 2018*). This suggests that elevation and/or humidity variables may have contributed to the differentiation of *D. belus*, possibly acting as a barrier between this species and other lowland species in the Agenor group. In addition, *D. belus* falls much less frequently in pitfall traps compared to *D. agenor,* even though it is abundant when manually collected in cattle dung pads; this may indicate the existence of differences in behaviour or at least in food preferences.

We recovered *D. yucatanus* and *D. parcepunctatus* as sister lineages but the total-evidence species delimitation analysis failed to recognise them as different species despite they belonging to different species-groups (Inachus and Batesi) and having a very distinct geographic distribution. This finding is consistent with a previous molecular phylogeny built for the tribe Scarabaeidae that recovered *D. yucatanus* and *D. parcepunctatus* as sister species across all the 9008 ML trees sampled (*Borrow, 2011*). Unfortunately, the existing sampling and information on these species is insufficient to explain this pattern and more studies about the ecology and/or distribution of these species are needed.

Our data strongly supported the existence of the species-groups Mamillatus, Mormon and Boreus, and overall, this grouping coincides with differences in elevational distribution. For example, aedeagus morphology grouped the species-groups Mamillatus and Boreus in a single cluster that contains only lowland species with Amazonian distribution (green in Fig. 1), while the Mormon group is composed only by highland species restricted to the Andes (blue in Fig. 1). The molecular phylogeny separated the lowland species in the corresponding Mamillatus and Boreus groups, but these were not reciprocally monophyletic since the Boreus group is more closely related to the highland species. Although *D. podalirius* and *D. boreus* showed phylogenetic divergence (Fig. 2), which can be partially explained by the ability of *D. boreus* to reach higher elevations (100–1,000 masl) than *D. podalirius* (100–350 msal) in the foothills of the Eastern Cordillera of Colombia (*Medina et al., 2001*), the total-evidence species delimitation failed to recover them as independent species, which may suggest they are different populations of a single species.

Species in the Mormon species-group clustered all together and were hardly distinguishable at the molecular level. Even so, *D. satanas* split in two monophyletic

clusters that correspond to Central American and Colombian individuals, suggesting they are different entities. Nonetheless, the species delimitation method applied was not able to discriminate these taxa as independent (except for *D. quinquelobatus*). Interestingly, while all species in the Mormon group are found in elevations between 1,000 and 2,000 masl, only *D. quinquelobatus* goes down and reaches the foothills of the Eastern Colombia Cordillera (120–2,200 masl (*Sarmiento-Garcés & Amat-García, 2014*)), thus receiving some influence from the Orinoquia and Amazonia. Our phylogeny suggests that the highland clade derives from lowland species, although this needs further confirmation.

Additionally, while Colombian *D. satanas* showed population structure associated with the Andean Cordilleras, and under morphological based taxonomic studies these populations would be identified as two species, none of our delimitation analyses discriminated these populations as separate entities. Therefore, the currently available data indicates that Colombian *D. satanas* is a single species that displays a remarkable phenotypic variation in the number of protuberances (two and four) on the pronotum of females. This is a unique condition in the Scarabaeinae subfamily, and this variation is associated with geography to some extent. At present it is not possible to pinpoint the factors contributing to the maintenance of this variation although processes such as sexual selection, known to drive horn polymorphism in multiple species of beetles (*Emlen, Corley Lavine & Ewen-Campen, 2007*; *Kijimoto et al., 2013*; *Simmons & Watson, 2010*), may be implicated. Also, the fact that the four-protuberances morph is collected only in open and disturbed habitats whilst the two-protuberances morph is mostly found in forested habitats suggests that variables such as temperature variation, vegetation coverage and/or food availability, that drastically differ between the two habitats, may be promoting the differentiation between these morphs, at least in females.

In general, the results of our total-evidence species delimitation analyses under different scenarios of population size and divergence time were remarkably congruent. However, when the delimitation analysis was based on molecular or morphological data alone the results were much more sensitive to the *priors* used, either supporting most the *a priori* morphospecies assignments (molecular data) or almost none at all (morphology data). This pattern has been previously observed in other studies of species delimitation in beetles, where only the combination of morphological and molecular data resulted in robust estimates by reducing the sensitivity to *prior* parameter choice (*Eberle, Warnock & Ahrens, 2016*). Our current sampling (in terms of taxa and genes) does not permit us to make definite interpretations on the validity of all species of *Dichotomius*, but we can confidently recognise *D. nisus, D. belus* and *D. mamillatus* as valid and well differentiated species. Although it would have been ideal to reach a final conclusion for all species evaluated here, species delimitation methods are extremely sensitive to multiple biases such as insufficient or unbalanced sampling, incomplete lineage sorting, population structure and/or hybridisation (*Astrin et al., 2012*; *Carstens et al., 2013*; *Meyer & Paulay, 2005*; *Petit & Excoffier, 2009*; *Sukumaran & Knowles, 2017*; *Yang et al. 2019*). In our study, we used the morphology of male genitalia as diagnostic trait but other traits used for the identification of *Dichotomius* (*Nunes, 2017*) need to be considered. Also, we had an unbalanced representation of species in our dataset, which also needs to be corrected

in future studies. Despite these limitations, this is the first time an integrative species delimitation approach is implemented in *Dichotomius* and we feel that our analytical procedures were adequate enough to reveal the ambiguous taxonomic position of several taxa.

Altogether, our findings indicate the need to revise the current taxonomic classification of *Dichotomius* in the light of both morphological and molecular data. Only such an integrative approach will allow a comprehensive characterisation of the diversity, ecology and distribution of species in this genus, to ultimately understand the mechanisms and processes involved in their adaptation, diversification and speciation.

## CONCLUSIONS

*Dichotomius* is a rich and diverse dung beetle genus (*Nunes & Vaz-de-Mello, 2016*) that belongs to the tribe Dichotomini, one of the most problematic tribes in Scarabaeinae (*Tarasov & Dimitrov, 2016*). Therefore, the validation of its taxonomy and evolutionary relations constitutes a step towards a reassessment of the systematic and phylogenetics of New World dung beetles as a whole. Our implementation of a total-evidence species delimitation approach that integrates genetic and phenotypic information provided a powerful tool to accurately delineate lineages in *Dichotomius* and suggest the existence of fewer species in the genus. We recommend including additional species as well as to sample more loci and phenotypic traits to further improve the taxonomy and biogeography of *Dichotomius.* However, we highlight the importance of our findings in the understanding of the biogeographical and evolutionary processes influencing this genus, as well as their significance for taxonomy and conservation.

## ACKNOWLEDGEMENTS

We would like to thank Santiago Montoya for helping with the identification of some species and providing valuable opinions. We also thank Camila Ruiz for her help processing some samples of *D. satanas*. All specimens used came from the following collections: CALT-ECC (Colombian Collection ID 2), MUD (Colombian Collection ID 46), and CAUR (Colombian Collection ID 229).

### Funding

This work was funded by the FPIT (Banco de la República—Colombia) grant number 3584 and the Dirección de Investigación e Innovación (Universidad del Rosario—Colombia) grant number IV-TSE009. The funders had no role in study design, data collection and analysis, decision to publish, or preparation of the manuscript.

### Grant Disclosures

The following grant information was disclosed by the authors:
FPIT (Banco de la República—Colombia): 3584.

Dirección de Investigación e Innovación (Universidad del Rosario—Colombia): IV-TSE009.

## Competing Interests

The authors declare there are no competing interests.

## Author Contributions

- Carolina Pardo-Diaz and Camilo Salazar conceived and designed the experiments, performed the experiments, analyzed the data, contributed reagents/materials/analysis tools, prepared figures and/or tables, authored or reviewed drafts of the paper, approved the final draft.
- Alejandro Lopera Toro conceived and designed the experiments, contributed reagents/materials/analysis tools, approved the final draft.
- Sergio Andrés Peña Tovar performed the experiments, contributed reagents/materials/-analysis tools, approved the final draft.
- Rodrigo Sarmiento-Garcés contributed reagents/materials/analysis tools, approved the final draft.
- Melissa Sanchez Herrera performed the experiments, analyzed the data, contributed reagents/materials/analysis tools, approved the final draft.

## Data Availability

Sequences are available in GenBank: MK614968–MK615034 (3′ COI), MK628601–MK628681 (5′ COI) and MK614942–MK614967 (28S). Morphology data is available in the Supplemental File.

## Supplemental Information

Supplemental information for this article can be found online at http://dx.doi.org/10.7717/peerj.7332#supplemental-information.

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
