# Peer review of "Taxonomic reassessment of the genus Dichotomius (Coleoptera: Scarabaeinae) through integrative taxonomy"

_PeerJ, doi:10.7717/peerj.7332_

## Round 0.1 · original submission · Major Revisions

Dear Dr. Pardo-Diaz and colleagues:

Thanks for submitting your manuscript to PeerJ. I have now received two independent reviews of your work, and as you will see, the reviewers raised some concerns about the research. Despite this, these reviewers are optimistic about your work and the potential impact it will lend to research on Dichotomius taxonomy and systematics. Thus, I encourage you to revise your manuscript accordingly, taking into account all of the concerns raised by both reviewers.

While the concerns of the reviewers are relatively minor, this is a major revision to ensure that the original reviewers have a chance to evaluate your responses to their concerns.

Please note that reviewer 1 has kindly provided a marked-up version of your manuscript. These concerns, as well as those listed in the general reviews should all be adequately addressed in your revision. Overall, the presentation of your methods and results needs to be improved, and additional methodological analyses stand to greatly improve your work.

I look forward to seeing your revision, and thanks again for submitting your work to PeerJ.

Good luck with your revision,

-joe

Reviewer 1 ·

Basic reporting

The authors do a nice job of outlining the reason why this phylogenetic study was done. They have sufficiently cited past studies, and have justified their choice of data and taxon selection.

Experimental design

1. When introducing the 16S marker in the materials and methods, please indicate the number of species for which this marker was available.
2. They should be very explicit and clear in the methods to explain why they ran haplotype networks on the 3' and 5' COI ( line 213/214) separately.
3. Symbols seem to be missing in line 225.
4. The figure 2 should include an image of the genitalia measured.
5. The phylogenetic data is presented in a very clear and concise way.

Validity of the findings

This phylogeny is an excellent first step in evaluating this genus. The data is robust, and uses several independent sources that are largely congruent.

Additional comments

1. Some words are italicized that need not be, line 29 on page 1, " richest within the New World Scarabaeinae". Line 93/94 has extra set of parathenses: "(although this result is based on a small number of
2. 93 species of Dichotomius and only one individual per species, Bocak et al. 2014; Monaghan et al.
94 2007)."
3. Lastly, the references are not uniform in the application of the "-" and "en dash" between page numbers.

·

Basic reporting

My review doesn't exactly cover these items, but I see no issues in this regard.

Experimental design

Largely good. My detailed comments (in the 'General Comments' box) describe concerns about how some of the data were parsed, analyzed, and interpreted.

Validity of the findings

Overall I believe the conclusions are largely justified. However, some of their results could be presented more clearly, and some recommended reanalyses might alter the conclusions in small ways. More comments below.

Additional comments

This paper evaluating (very preliminarily) the phylogeny of the large Neotropical genus Dichotomius has a strong introduction. The authors do a good job of creating a good story from a group of beetles that has received relatively little press. The combination of nuclear and mitochondrial molecular data and morphometric data on genitalia allows for some generally well executed and synthetic analyses addressing species level phylogeny, some phylogeographic questions, and species delimitation. So I think most of the ingredients are in place for an excellent paper. However, I do have a number of concerns, mainly with the presentation, which can be hard to follow in places, and with their overpartitioning their data, treating two contiguous fragments of COI as if they were independent markers. The authors also seem to misinterpret the purpose and results of 'species delimitation' analysis, since most species included aren't (apparently) represented by multiple exemplars. So I would recommend some reanalyses, a fair bit of figure reformatting, and considerable text revision to address these concerns, detailed further below. In addition, my annotated copy of the pdf includes a number of additional minor edits, comments, and questions that should be addressed.

There's no justification (nor any presented) for treating the two halves of COI separately in the quantitative and phylogenetic analyses. These sequences should be concatenated and treated as coherent haplotypes (albeit with gaps at the primer junction). Furthermore, I don't understand why so much of the COI data was excluded for higher level phylogenetic analyses. Were the authors concerned about missing data? They don't state this as a their justification, and shouldn't assume that it would be a problem without attempting some analyses. I would think that having the largest numbers of individuals included that are available for the best represented marker would be very valuable for obtaining clearer resolution among the species.

Through the figures and text, it's difficult to keep track of how many specimens represent each species in each analysis, and it's therefore difficult to interpret various statements about the monophyly and coherence of species. 79 of 95 individuals represent D. satanas, and another 52 came from GenBank. Without going into the supplementary tables one can't easily tell how these other specimens are distributed across the tree. So when the authors claim that iBPP was done for 'all species and D. satanas', we can't really tell how good a test of these other species coherence this represents.

The species delimitation figure places posterior probability boxes on branches below multispecies clades. How are these to be interpreted? Not as species probabilities, but as clade probabilities? If this is intended as a true species delimitation analysis, these should only be applicable to terminal clades with species represented by more than a single individual for molecular data. It is not clear to me that this is true for any species other than D. satanas.

Other miscellaneous points:

The figures use elaborate color schemes to convey information on classification, distribution, and natural history. These seem overly complicated, and are very hard to interpret, as many colors are very similar, and the significance of some of the habitat distinctions is unclear/questionable.

The phylogeographic results for satanas requires a map to put it into some kind of explicitly geographic context. It means relatively little any reader unfamiliar with Colombian geography.

Rooting the satanas tree with D. agenor seems a strange choice, and may result in a spurious rooting. Why not use one or more closer relatives?

The discussion goes into quite a bit of detail with regard to internal relationships and classification. But with only 16 of 171 species represented by DNA data here, it's hard to take these very seriously. Probably a lot of that could be compressed and made a bit more tentative.

---

## Round 0.2 · Minor Revisions

Dear Dr. Pardo-Diaz and colleagues:

Thanks for re-submitting your manuscript to PeerJ. I have now received one review of your work, and as you will see, the reviewer still has some concerns about the manuscript. Thus, I encourage you to revise your manuscript accordingly, taking into account all of the new concerns raised by the reviewer.

Please note that the reviewer has once again kindly provided a marked-up version of your manuscript. These concerns, as well as those listed in the general reviews should all be adequately addressed in your revision.

I look forward to seeing your revision, and anticipate accepting your work for publication once these minor issues are addressed.

Thanks again for submitting your work to PeerJ.

Good luck with your revision,

-joe

·

Basic reporting

See general comments..

Experimental design

No comment

Validity of the findings

No comment

Additional comments

While the changes the authors have made to their analyses represent a significant improvement, the paper is still rough in interpretation and presentation. Please see my submitted, tracked-changes version for more (not exhaustive) details. Some of the main issues have to do with discussion of groups (subgenera and others) found to be non-monophyletic. The authors don't explain the findings very clearly, and seem almost to contradict their results in a couple places. Most of my other comments are quite minor, but for a second revision, this is still a little rough.

---

## Round 0.3 · accepted · Accept

Dear Dr. Pardo-Diaz and colleagues:

Thanks for revising your manuscript. The sole re-reviewer is very satisfied with your revision (as am I). Great! There are 2 minor typos identified by the reviewer in the attached. These can be addressed while in production.

Best,

-joe

·

Basic reporting

see general comments

Experimental design

see general comments

Validity of the findings

see general comments

Additional comments

This version looks clearer. I note just a couple spelling changes in the attached (one a verb tense mismatch, one just a misspelling).